# Analysis and Design of New Actuator Used for Full-Wide Screen LCD

**Jun-Hyung Kim, Yuan-Wu Jiang and Sang-Moon Hwang \***

School of Mechanical Engineering, Pusan National University, Busan 609-735, Korea;
joonyng7@gmail.com (J.-H.K.); evan.jiang.pnu@gmail.com (Y.-W.J.)
**\*** Correspondence: shwang@pusan.ac.kr; Tel.: +82-051-510-3204

**Abstract:** The design of the smartphone LCD is gradually shifting to a full-wide screen. This requires a new type of actuator rather than a conventional dynamic receiver. For this purpose, a new actuator is used as a dynamic receiver. This paper introduces and analyzes new types of actuators and compares their force factors. For an analysis tool, a prototype of actuator is analyzed and verified with its experimental results. By using this verified tool, new types of actuators are designed and analyzed. Furthermore, a comparison is done to find out which one is the best for vibration performance.

**Keywords:** actuator; structure design; coupled analysis

## 1. Introduction

Actuators are used mainly to generate vibration in mobile devices. Actuators mainly perform vibration-related functions, such as vibration and sound generation. These functions are in fact a very important medium that connects humans and machines, and it becomes more important as time goes by. These days, consumers are actively demanding not only visual stimuli through the screen and auditory stimuli through the speaker, but also tactile stimuli through the actuators. For this reason, research on the actuators is actively progressing step by step.

In previous research, a novel design of solenoid-type vibrators using electromagnetic and mechanical was introduced [1,2]. A coin shaped linear vibrator with a conical spring was proposed to achieve good vibration performance [3]. Attila et al. proposed a new linear electromagnetic actuator that made use of a metal spring [4]. A horizontal, linear vibrating actuator to reduce thickness of the smartphone was analyzed and developed [5,6]. Nam et al. proposed a novel design of a linear actuator with a large magnetic force to reduce the response time. The response time to reach the steady state of vibration was investigated through the equivalent mass–spring–damper system of the linear actuator [7]. Nam et al. developed a resonant piezoelectric vibrator with a high displacement output at a low haptic frequency [8]. Nam et al. developed and proposed an externally leveraged circular resonant piezoelectric actuator with haptic natural frequency and a fast response time. [9]. A vibration actuator was developed based on active magnetic springs, which means there is no spring structure in the mechanical components [10]. Coupled analysis for the vibration motor using an electromagnetic and mechanical finite element method (FEM) was introduced and verified through an experiment [11].

Smartphones are the most used electronic devices in the world and the design of the smartphone LCD is gradually shifting to full-wide screen. By deleting the home button located at the bottom of the LCD, the screen was able to get bigger; however, the dynamic receiver at the top of the smartphone LCD needs a hole for spreading sound, so the size of the smartphone LCD could not be increased if the dynamic receiver is used. The dynamic receiver is the term that refers to the micro speaker. The change of the smartphone appearance is shown in Figure 1.

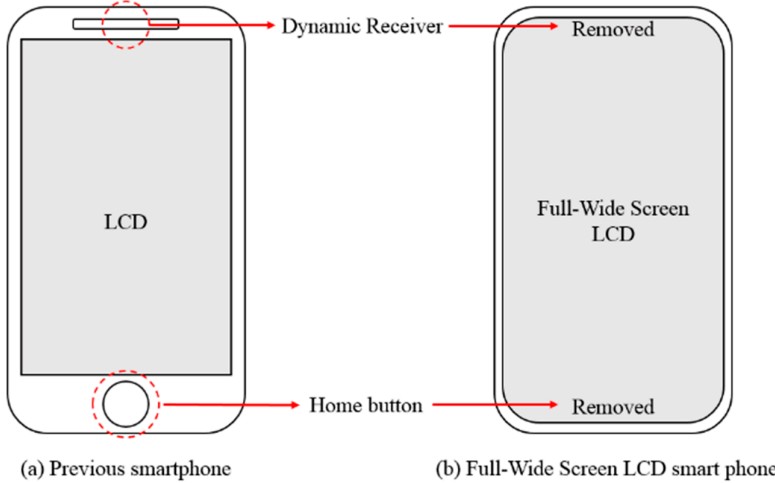

(a) Previous smartphone　　　　　　　　(b) Full-Wide Screen LCD smart phone

**Figure 1.** Change of smartphone design.

Thus, a method of attaching an actuator to a full-wide screen smartphone frame to generate sound was designed. Compared to traditional dynamic receivers, which generate the sound by vibrating the coil and the diaphragm directly, the new method creates sound by attaching the actuator to the frame of smartphones and generating and transmitting the vibrations of the actuators to the smartphone frame. The difference in the sound generation principle between dynamic receiver and actuator is briefly shown in Figure 2.

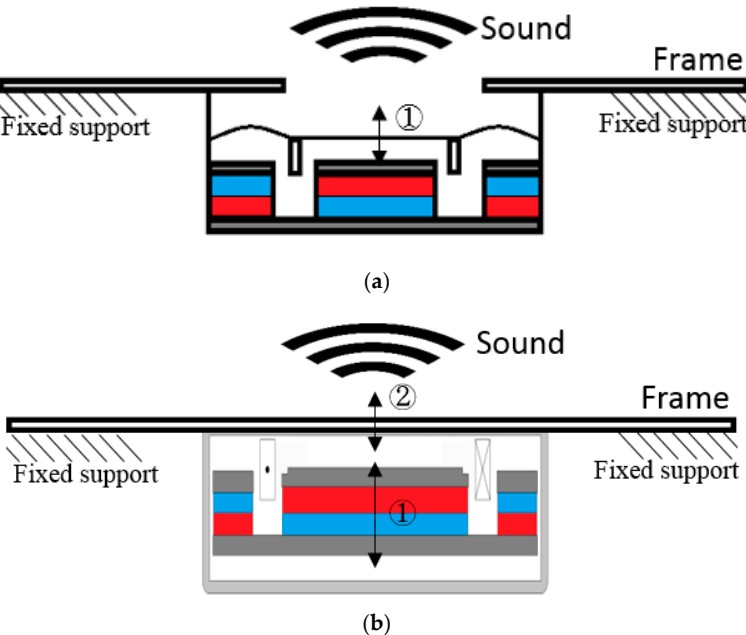

**Figure 2.** (**a**) Sound generation principle of the dynamic receiver; (**b**) Sound generation principle of the actuator.

However, because of the lack of vibration performance, the present actuator is not suitable for the dynamic receiver. To use the actuator as a dynamic receiver, the new actuator with high vibration performance is needed. In the present work, the actuator with high vibration performance is designed and analyzed. By using an electromagnetic–mechanical coupling simulation method, considering nonlinear parameters, the performance of the actuator was analyzed. Based on electromagnetic–mechanical coupled analysis, a prototype actuator which is 3 MG was created and is shown in Figure

3. A prototype actuator has the input power of 1 W, outer dimensions of 13 × 10 × 2.65 mm, and a DCR as 8 ohm. Input voltage is a sinusoidal wave of 2.828 V. By using a prototype actuator, the E–M coupled analysis was verified by an experiment. Based on a proven analytical method, new actuators were designed and analyzed.

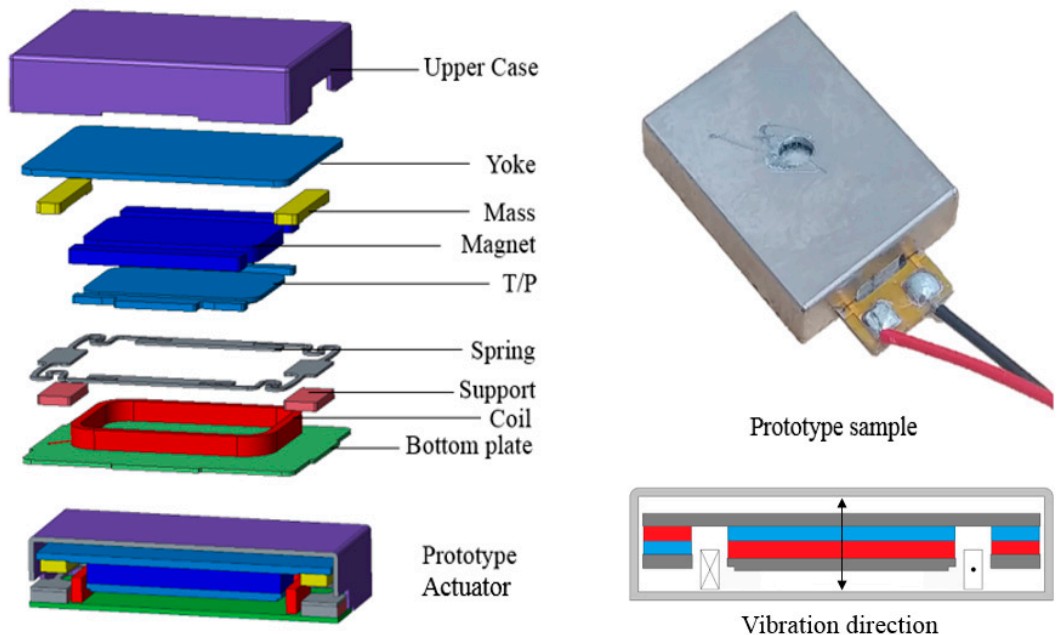

**Figure 3.** Prototype (3 MG) actuator structure.

## 2. Analysis Method and Experiment Verification

### 2.1. Electromagnetic Analysis

In the electromagnetic analysis, the inductance, speedance, and force factors are nonlinear parameters used for solving the voltage equation. Firstly, the flux density that exists inside a coil was obtained using the finite element method (FEM). By using flux density, flux linkage was calculated. The flux linkage is the product of the flux density, coil turns, and the coil sectional area. After calculating the flux linkage, inductance and speedance can be calculated by following this calculation method: inductance is defined as the derivative of the flux linkage to the current and speedance is defined as the derivative of the flux linkage to the displacement.

By using Lorentz force, the force factor on the coil is calculated. The force factor can be obtained by dividing the current applied to the Lorenz force. These parameters are involved in the following voltage equation:

$$V = iR + L(y,i)\frac{di}{dt} + K_{emf}(y,i)\frac{dy}{dt} \tag{1}$$

where $V$, $R$, $L(y,i)$, $K_{emf}(y,i)$, and $t$ are voltage, resistance, inductance, speedance and time, respectively. Those nonlinear parameters are used in the electromagnetic mechanical coupled analysis, and the nonlinear parameters are shown in Figure 4.

### 2.2. Mechanical Analysis

In the mechanical analysis, the displacement of the vibrating part was calculated using the FEM with forced vibration. The governing equation can be written as follows:

$$[M]\ddot{y} + [C]\dot{y} + [K]y = \{f_{Lorentz}\} \tag{2}$$

where [*M*], [*C*], and [*K*] indicate the mass, damping, stiffness, and y, {$f_{Lorentz}$} are displacement and the vector form of the Lorentz force that is obtained in the electromagnetic analysis.

By solving the governing equation, a transfer function, which is displacement per unit force in frequency domain, can be obtained by using a modal and harmonic analysis. The transfer function will be used in E–M coupled analysis. This is shown in Figure 4. The material properties used in the FEM are shown in Table 1.

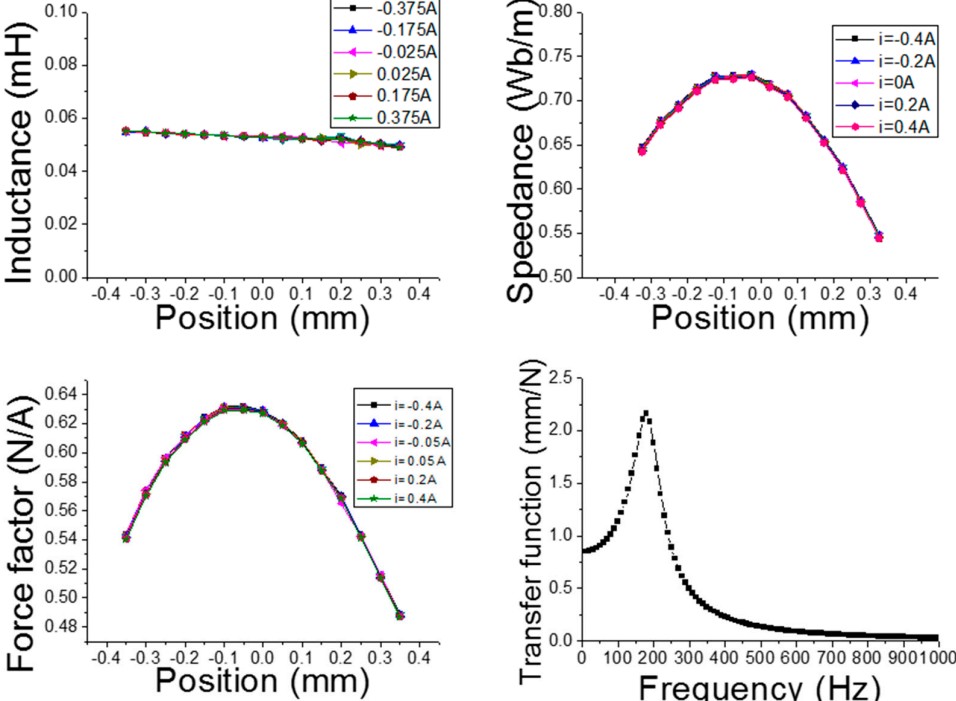

**Figure 4.** Nonlinear parameters and transfer function of the actuator.

**Table 1.** Material properties.

| Part | Material | Density (kg/m³) | Young's Modulus (GPa) | Poisson's Ratio |
|------|----------|-----------------|------------------------|------------------|
| Case | SUS301 | 7930 | 189 | 0.265 |
| Yoke | SPCC | 7830 | 207 | 0.290 |
| Mag | N50H | 7010 | 41.4 | 0.300 |
| Top plate | SPCC | 7830 | 207 | 0.290 |
| Spring | SUS301 | 7930 | 189 | 0.265 |
| Mass | Tungsten | 19,250 | 411 | 0.280 |

*2.3. Electromagnetic–Mechanical Coupled Analysis*

Input voltage for coupled analysis is applied with a sinusoidal wave of 2.828 V, which is done in the same condition as in the experiment. The coupled analysis and experiment have the same 1 W input power condition.

To do the coupled analysis, two different equations, which are the voltage equation from the electromagnetic domain and the forced vibration governing equation from the mechanical domain, need to be solved at the same time. These two equations can be expressed as follows:

$$V = iR + L(y,i)\frac{di}{dt} + K_{emf}(y,i)\frac{dy}{dt} = iR + L(y,i)\frac{di}{dt} + K_{emf}(y,i)\frac{K_f(y,i)i}{Z_{ms}} \tag{3}$$

$$[M]\ddot{y} + [C]\dot{y} + [K]y = \{f_{Lorentz}\} = K_f(y,i)i \tag{4}$$

where $K_f(y,i)$, $Z_{ms}$ is the force factor and mechanical impedance. Equation (3) is derived from Equation (1) in the electromagnetic analysis, and Equation (4) is derived from Equation (2) in the mechanical analysis. An important fact to take note of is that these two equations are connected by the force factor, and these equations affect each other, which means that these are coupled. This is the main reason E–M coupled analysis is needed for the actuator analysis. Without E–M coupled analysis, it is hard to obtain accurate simulation results.

To solve these equations simultaneously, a numerical iteration method, especially the numerical fixed-point iteration method, was used. By using this procedure, the forced vibration governing equation and the voltage equation can be solved at the same time, and then the simulation displacement and impedance can be obtained. A flow chart of E–M coupled analysis is shown in Figure 5. The simulation results were compared with the experimental results.

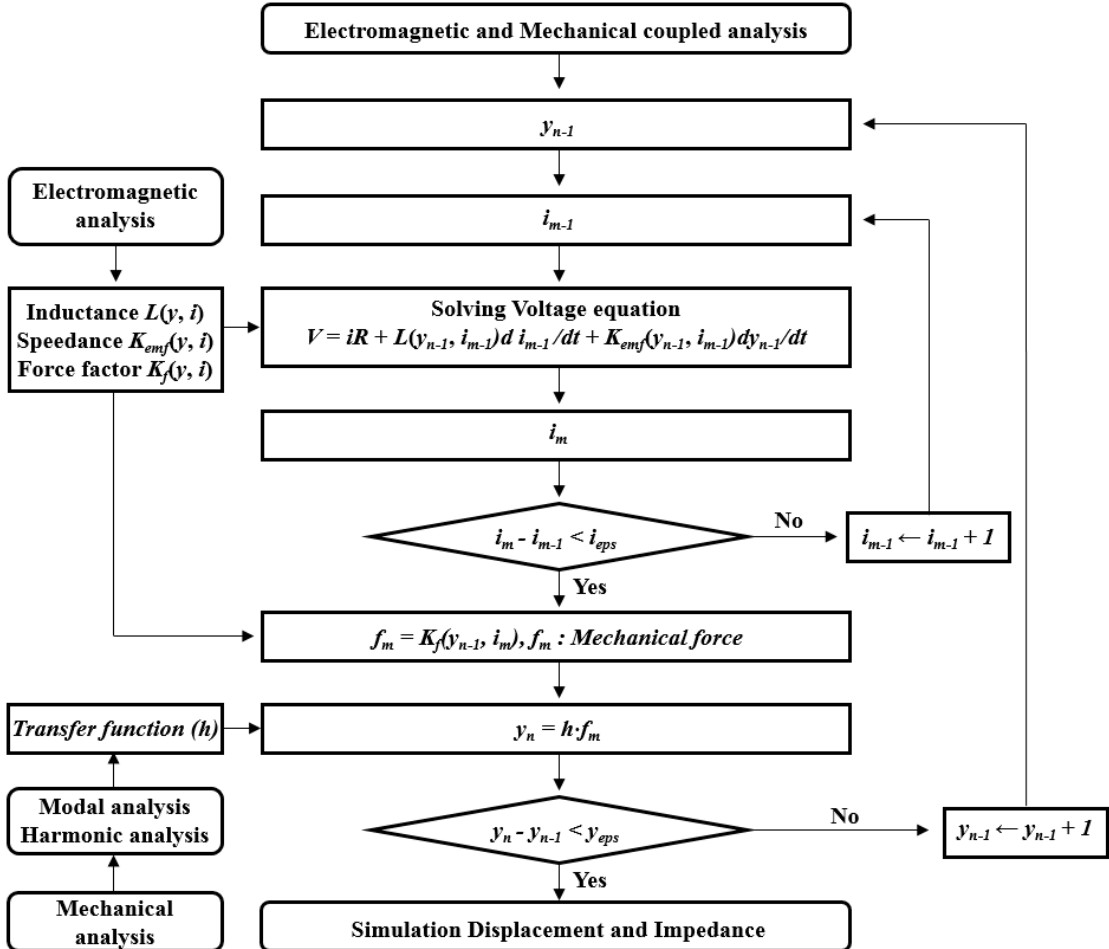

**Figure 5.** Flow chart of electromagnetic–mechanical coupled analysis.

*2.4. Experiment and Comparison*

The displacement and impedance of the actuator was measured by Klippel equipment that uses a laser. The method for measuring impedance is as follows: After applying the voltage into the actuator, the current depending on the frequency is measured by a current sensor in the Klippel equipment. By doing this, impedance is calculated by dividing the measured current into the applied voltage. The Klippel equipment is shown in Figure 6, and the simulation and experimental results

are shown in Figure 7; these are well matched. This means that the E–M coupled analysis used in this study can be used for the actuator design. By using the same method, the new actuator was designed and analyzed by comparing the force factor of the new actuator with the previous 3 MG type actuator.

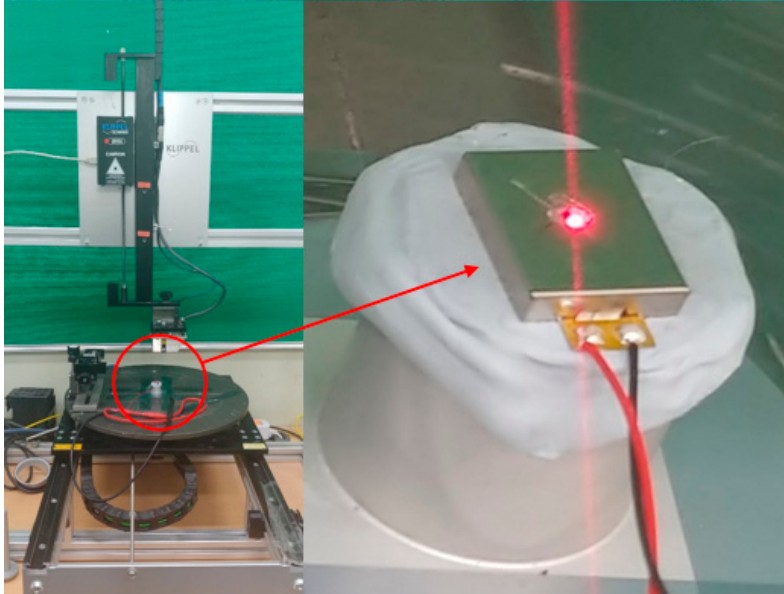

**Figure 6.** Klippel equipment.

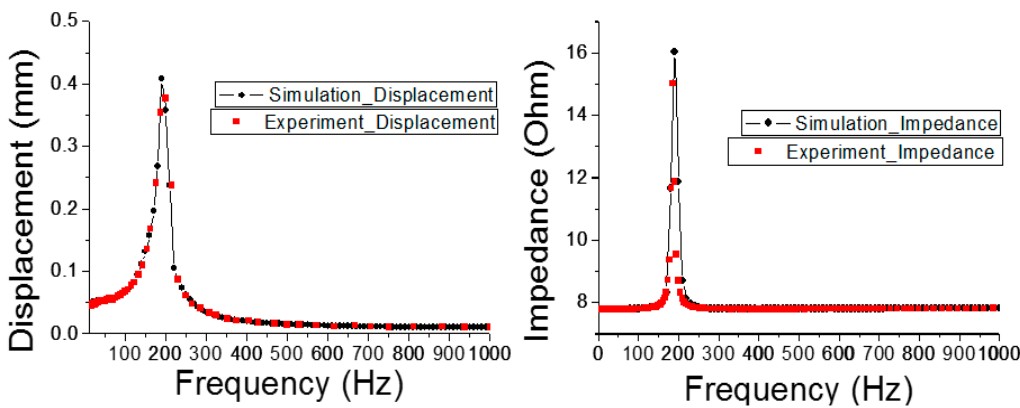

**Figure 7.** Comparison between simulation and experiment results.

## 3. Structure Design

The numbers 3 MG, 2 MG, and 5 MG refer to the magnets used in the actuator. A prototype actuator has three magnets, a 2 MG type actuator has two magnets, and a 5 MG type actuator has five magnets.

The prototype actuator has a low force factor due to the limitation of a 3 MG type. Since a 3 MG type actuator does not have a magnet-top plate structure on the long axis, the coil is barely active at that part, and vibration power is weaker than the other 2 MG or 5 MG type actuators. By changing the spring-support connection structure, the new actuator with a higher force factor is needed and two new actuator structures are generated.

A prototype actuator has the connections between the support from the bottom plate and the spring, and between the spring and the top plate (T/P). In order to prevent contact between components when the actuator has maximum displacement, the part where the mass part is currently attached must be made thin, because it must be kept at a constant distance from the spring support

structure. This part is too thin to apply the magnet-top plate structure, and if applied, it will become thick and cause a touching problem. Due to the touching problem, the prototype actuator's structure cannot be changed into 2 MG or 5 MG type actuators, due to the spring support connection structure.

To design a 2 MG or a 5 MG type actuator structure, the spring connection design is changed. A new actuator has the connections between the support from the upper case and the spring, and between the spring and the yoke. Because of these changes, it is possible to make 2 MG or 5 MG type actuators that have the advantage of generating higher force factors than the previous 3 MG type actuator.

Two new actuator structures are proposed and shown in Figure 8. These are unlike the 3 MG prototype and are 2 MG type and 5 MG type actuators with the same input power (1 W), outer dimensions (13 × 10 × 2.65 mm), and DCR (8 ohm) as the prototype.

Due to these changes, the new actuators are expected to have higher force factors, which will increase the vibration force.

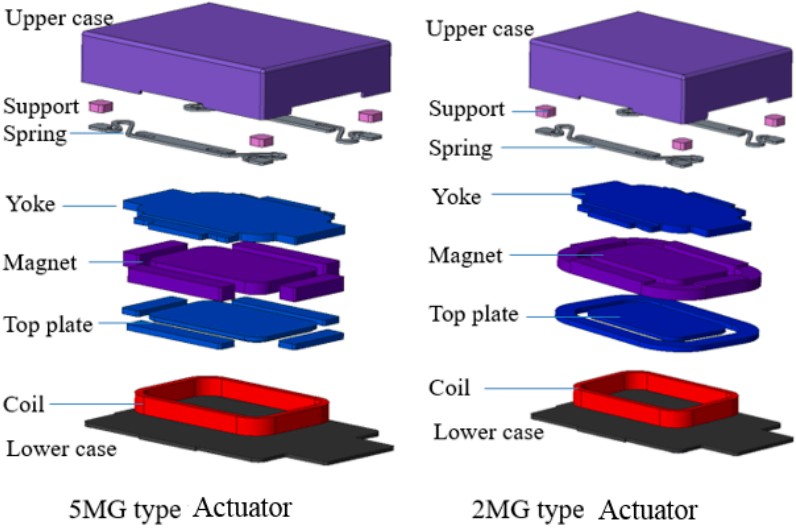

**Figure 8.** Structure of 5 MG and 2 MG type actuator.

## 4. Results

In the electromagnetic analysis, nonlinear parameters such as inductance, speedance, and force factor are achieved. Figure 9 shows a comparison between the prototype force factor and 5 MG and 2 MG type actuators. It shows that a 2 MG type actuator has the highest force factor, which has 44% higher value than the prototype. The vibration displacement of the actuators is mainly affected by the force factor. Therefore, it is expected that the 2 MG type actuator will have the highest vibration performance.

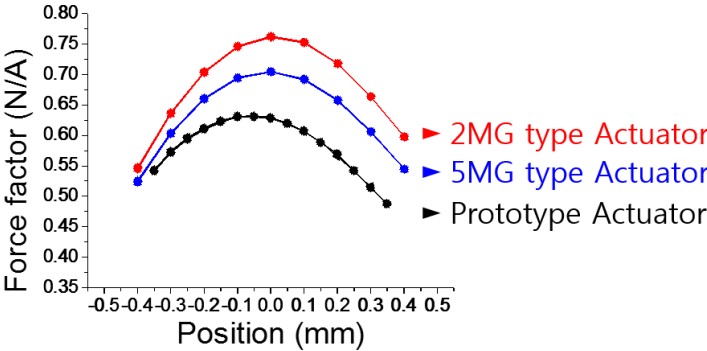

**Figure 9.** Force factor comparison between three actuators.

After finishing the electromagnetic coupled analysis, the vibration displacements of the three actuators are obtained and the result is the same as the expectation. The 2 MG type actuator has 40% higher vibration displacement than the prototype 3 MG actuator. The vibration displacement comparison results are demonstrated in Figure 10.

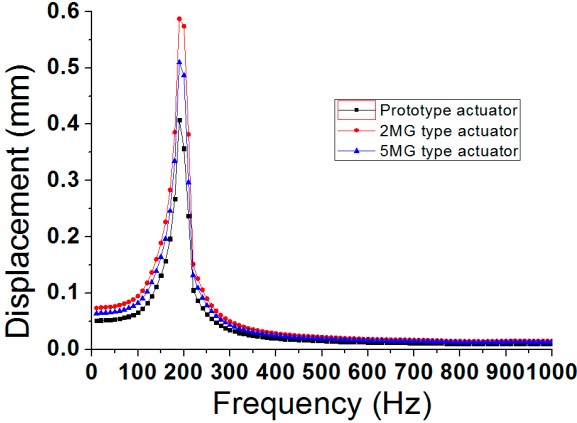

**Figure 10.** Vibration displacement comparison between three actuators.

## 5. Conclusions

In this study, two new types of actuators were introduced and analyzed. The analysis method is based on the convergence of the numerical iteration method, which uses electrical and magnetic FEM and mechanical FEM.

This analysis method was verified by a comparison between the experimental and simulation results of the prototype actuator and it is well matched.

By using the same analysis method used for the prototype actuator, the vibration displacements of three actuators were obtained and compared.

By using the verified method, it is possible to select the actuator with the best vibrating performance. According to the results, the 2 MG type actuator has the best vibrating performance. For future research, the 2 MG type actuator will be manufactured, and its displacement will be achieved by experiment.

**Author Contributions:** Conceptualization, J.-H.K. and Y.-W.J.; methodology, J.-H.K.; software, J.-H.K.; validation, J.-H.K.; formal analysis, J.-H.K.; investigation, J.-H.K.; resources, J.-H.K. and Y.-W.J.; data curation, J.-H.K.; writing—original draft preparation, J.-H.K.; writing—review and editing, J.-H.K. Y.-W.J. and S.-M.H.; visualization, J.-H.K.; supervision, S.-M.H.; project administration, S.-M.H.

**Funding:** This research received no external funding.

**Conflicts of Interest:** The authors declare no conflict of interest.

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
