# Peer review of "Analysis and Design of New Actuator Used for Full-Wide Screen LCD"

_applsci, doi:10.3390/app9214599_

Round 1
Reviewer 1 Report
Hi,
Please find the comments and suggestions in the attached file.
Thanks,

Author Response
Thank you for your kind review.
Please see the attachment.

Reviewer 2 Report
The work deals with the current, important issues, in which, the Authors has an extensive experience. The paper is interesting. It is concise, but in some parts the description is too laconic. Despite this it makes overall positive impression. Some problems are found, that need to be resolved prior to publication in Applied Sciences journal. Not all the found issues are minor shortcomings. However, I believe that Authors can handle it.
lines 24-25 - something is missing here - "using electromagnetic and mechanical" but what exactly?
10 sources were cited. None of them is newer than two years. Are these all known recent developments in that field? All the references were cited in another publication of the Authors: "Analysis of a Vibrating Motor Considering Electrical, Magnetic, and Mechanical Coupling Effect". That paper seems to be in a similar field of research, but was not cited in this paper. Is not the reviewed manuscript a continuation of that works? Do not get it wrong. These are not objections, rather just doubts.
Sometimes smartphone is written together, sometimes separately. Authors are asked to uniform a spelling of this word.
Section 2 - Analysis method and experiment verification - Results with specific figures were presented but no input data was provided. This remark concerns both theoretical and experimental analysis. Nothing about dimensions, materials, voltages, inductances and so on. This is a serious fault.
line 123 - what are these limitations? Could you specify?
Section 3 - Abbreviations 3MG, 2MG and 5MG were not clarified. It can be concluded that this is about the number of magnets, but it is not quite obvious. This should be clearly exposed in the text. Especially since in Figures 3 and 8 the differences in the number of magnets were not marked, but should be. Explanation, why did you use 2, 3 and 5 magnet, but not, for example, 4, 6 or 7 will be appreciated too.
Regarding the presented results - the figures are better in one case in another worse, but how this all deals with an end-user acceptance. It is known that at this time experimental research are not possible in this matter, but is it possible to indicate factors that are crucial in this point of view? What do the Authors think about that?
In general, the article is valuable contribution to a development in mechatronic devices, but many things are unclear, which needs to be improved in order the manuscript to be published.
Author Response

(The authors gave the same response as above.)

Round 2
Reviewer 1 Report
Please notice the added texts in blue colour.

Author Response
Thank you for your comments.

Reviewer 2 Report
Authors have made appropriate effort in order to correct their paper. Any doubts have been clarified enough. Missing information has been added. The reviewer recommends the revised manuscript to be published in Applied sciences journal.
Author Response
Thank you for your comments.
